



# Mutual promotion effect between aerosol particle liquid water and nitrate formation lead to severe nitrate-dominated particulate matter pollution and low visibility

Yu Wang[1,2,a], Ying Chen[3,a], Zhijun Wu[1,4,5,*], Dongjie Shang[1], Yuxuan Bian[6], Zhuofei Du[1,b], Sebastian H. Schmitt[4,7,c], Rong Su[1,d], Georgios I. Gkatzelis[4,7,e,f], Patrick Schlag[4,7,g], Thorsten Hohaus[4,7], Aristeidis Voliotis[2], Keding Lu[1,4,5], Limin Zeng[1,4], Chunsheng Zhao[8], Rami Alfarra[2,9], Gordon McFiggans[2], Alfred Wiedensohler[10], Astrid Kiendler-Scharr[4,7], Yuanhang Zhang[1,4,5], Min Hu[1,4,5]

[1]State Key Joint Laboratory of Environmental Simulation and Pollution Control, College of Environmental Sciences and Engineering, Peking University, Beijing 100871, China
[2]Centre for Atmospheric Science, School of Earth and Environmental Sciences, The University of Manchester, Manchester M13 9PL, UK
[3]Lancaster Environment Centre, Lancaster University, Lancaster, LA1 4YQ, UK
[4]International Joint Laboratory for Regional Pollution Control, 52425 Jülich, Germany, and Beijing 100871, China
[5]Collaborative Innovation Center of Atmospheric Environment and Equipment Technology, Nanjing University of Information Science and Technology, Nanjing 210044, China
[6]State Key Laboratory of Severe Weather, Chinese Academy of Meteorological Sciences, Beijing, 100081, China
[7]Institute for Energy and Climate Research, IEK-8: Troposphere, Forschungszentrum Jülich, 52425 Jülich, Germany
[8]Department of Atmospheric and Oceanic Sciences, School of Physics, Peking University, Beijing 100871, China
[9]National Centre for Atmospheric Science, School of Earth and Environmental Sciences, The University of Manchester, Manchester, M13 9PL, UK
[10]Leibniz Institute for Tropospheric Research, 04318 Leipzig, Germany

[a]These authors contribute equally to this work
[b]Now at Center for Urban Transport Emission Research & State Environmental Protection Key Laboratory of Urban Ambient Air Particulate Matter Pollution Prevention and Control, College of Environmental Science and Engineering, Nankai University, Tianjin, 300071, China
[c]Now at TSI GmbH, 52068 Aachen, Germany
[d]Now at Guangdong Science and Technology Monitoring and Research Center, Guangzhou 510033, China
[e]Now at NOAA Earth Systems Research Laboratory, Boulder, Colorado 80305, United States
[f]Now at Cooperative Institute for Research in Environmental Sciences, Boulder, Colorado 80309, United States
[g]Now at Shimadzu Deutschland GmbH, 47269 Duisburg, Germany

*Correspondence to*: Zhijun Wu (zhijunwu@pku.edu.cn)





**Abstract.** As has been the case in North America and Western Europe, the $SO_2$ emissions substantially
reduced in North China Plain (NCP) in recent years. A dichotomy of reductions in $SO_2$ and $NO_x$
concentrations result in the frequent occurrences of nitrate ($pNO_3^-$)-dominated particulate matter
pollution over NCP. In this study, we observed a polluted episode with the nitrate mass fraction in non-
refractory $PM_1$ (NR-$PM_1$) up to 44% during wintertime in Beijing. Based on this typical $pNO_3^-$-
dominated haze event, the linkage between aerosol water uptake and $pNO_3^-$ formation, further
impacting on visibility degradation, have been investigated based on field observations and theoretical
calculations. During haze development, as ambient relative humidity (RH) increased from ~10% up to
70%, the aerosol particle liquid water increased from ~1 $\mu g/m^3$ at the beginning to ~75 $\mu g/m^3$ at the
fully-developed haze period. Without considering the water uptake, the particle surface area and the
volume concentrations increased by a factor of 4.1 and 4.8, respectively, during the development of
haze event. Taking water uptake into account, the wet particle surface area and volume concentrations
enhanced by a factor of 4.7 and 5.8, respectively. As a consequence, the hygroscopic growth of particles
facilitated the condensational loss of dinitrogen pentoxide ($N_2O_5$) and nitric acid ($HNO_3$) to particles
contributing $pNO_3^-$. From the beginning to the fully-developed haze, the condensational loss of $N_2O_5$
increased by a factor of 20 when only considering aerosol surface area and volume of dry particles,
while increasing by a factor of 25 considering extra surface area and volume due to water uptake.
Similarly, the condensational loss of $HNO_3$ increased by a factor of 2.7~2.9 and 3.1~3.5 for dry and wet
aerosol surface area and volume from the beginning to the fully-developed haze period. Above results
demonstrated that the $pNO_3^-$ formation is further enhanced by aerosol water uptake with elevated
ambient RH during haze development, in turn, facilitating the aerosol taking up water due to the



hygroscopicity of nitrate salt. Such mutual promotion effect between aerosol particle liquid water and

nitrate formation can rapidly degrade air quality and halve visibility within one day. Reduction of

nitrogen-containing gaseous precursors, e.g., by control of traffic emissions, is essential in mitigating

severe haze events in NCP.




# 1 Introduction

Aerosol particle hygroscopicity plays an important role in air quality deterioration and cloud formation (Yu, 2009;Fitzgerald, 1973;Kreidenweis and Asa-Awuku, 2014;Wang and Chen, 2019;McFiggans et al., 2006) and can also directly influence aerosol measurements (Chen et al., 2018a). In atmospheric environments influenced by anthropogenic activities, particulate secondary inorganic compounds are often dominated by ammonium sulfate ($(NH_4)_2SO_4$) and ammonium nitrate ($NH_4NO_3$) (Heintzenberg,

1989), which originate from the oxidation of sulfur dioxide ($SO_2$) and nitrogen oxides ($NO_x$) via well-established chemical pathways (Calvert et al., 1985). The abundance of secondary inorganic components is one of the most important factors determining particle hygroscopicity (Swietlicki et al., 2008), thereby governing the aerosol liquid water content under ambient moist conditions. Increased aerosol particle liquid water could accelerate secondary inorganic and organic aerosol formation by

decreasing the kinetic limitation of mass transfer of gaseous precursors and providing more medium for multiphase reactions (Mozurkewich and Calvert, 1988;Cheng et al., 2016;Wang et al., 2016;Ervens et al., 2011;Kolb et al., 2010).

Sulfuric acid ($H_2SO_4$) is formed from the oxidation of $SO_2$ via gaseous and multiphase reactions. $H_2SO_4$ is subsequently fully or partly neutralized by gaseous $NH_3$ taken up on particles, resulting in the

formation of $(NH_4)_2SO_4$ and / or $NH_4HSO_4$. Any remaining $NH_3$ is available to neutralize $HNO_3$ to form particulate $NH_4NO_3$ (Seinfeld. and Pandis., 2006) (and further excess $NH_3$ can neutralize any available HCl to form particulate $NH_4Cl$). Over the past several decades, substantial efforts have reduced emissions of both $SO_2$ and $NO_x$ improving the local and regional air quality all over the world.





For example, $SO_2$ and $NO_x$ emissions were reduced by 82% and 54% in the majority of European

Environment Agency member countries between 1990 and 2016 (https://www.eea.europa.eu/data-and-maps/indicators/main-anthropogenic-air-pollutant-emission s/assessment-4). In consequence, an

increasing trend of $NO_3^-/SO_4^{2-}$ molar ratio was observed in long-term measurements at Leipzig,

Germany (Spindler et al., 2004) and at some other European sites from the European Monitoring and

Evaluation Programme (EMEP) (Putaud et al., 2004). In recent years, China has also managed to reduce

$SO_2$ emissions by 75% since 2007 (Li et al., 2017a), whereas $NO_x$ emissions declined only by ~10%

between 2011 and 2015 (de Foy et al., 2016). Similar with European countries, the dominant inorganic

component in fine aerosol particles has switched from sulfate to nitrate in the recent years (Sun et al.,

2015;Hu et al., 2017;Hu et al., 2016;Wu et al., 2018;Guo et al., 2014;Huang et al., 2014;Huang et al.,

2010;Ge et al., 2017;Xu et al., 2019a;Xie et al., 2019;Li et al., 2018). Field measurements show that

annually averaged $NO_3^-/SO_4^{2-}$ molar ratio of NR-PM$_1$ (non-refractory PM$_1$) in 2012 (1.3~1.8) (Sun et

al., 2015) has significantly increased compared to that in 2008 (0.9~1.5) (Zhang et al., 2013).

Comparably, the $NO_3^-/SO_4^{2-}$ molar ratio of PM$_{2.5}$ increased substantially, from 1.5 before 2013 to 3.33

in 2017 (Xu et al., 2019a). Model simulations have also shown that the simulated annual mass

concentration of nitrate and its mass fraction in secondary inorganic components over North China

increased by 17~19% and 7% respectively, while the sulfate mass and fraction decreased by 10~19%

and 6% between 2006 and 2015 under the assumption of constant $NH_3$ emissions (Wang et al., 2013).

However, $NH_3$ emissions have been observed by satellites to increase by ~30% from 2008 to 2016 over

the North China Plain (NCP) (Liu et al., 2018), further increasing the potential for nitrate formation

(Wang et al., 2013).

Over the NCP region, heavy haze events are typically associated with enhanced ambient RH levels. This leads to an increased aerosol liquid water content (Wu et al., 2018), which will influence the particulate nitrate formation by changing the reactive uptake of precursors and the thermodynamic equilibrium of ammonium nitrate (Cheng et al., 2016;Wang et al., 2016;Wang et al., 2017;Yun et al., 2018;Yue et al., 2019). To date, few studies reported aerosol liquid water content over NCP region

(Wang et al., 2018;Bian et al., 2014;Cheng et al., 2016;Wu et al., 2018). However, the observational and theoretical analysis of the relationship between particulate nitrate formation and associated liquid water during haze events in China has been infrequently reported (Wu et al., 2018).

In this study, a self-amplification effect between particulate nitrate and liquid water is demonstrated by examining a nitrate-dominated fine particle Beijing pollution episode. The facilitation of particulate

nitrate formation by abundant liquid water is subsequently theoretically explored through the impacts of liquid water on thermodynamic equilibrium and heterogeneous reactions. Finally, the corresponding impacts on light extinction coefficient, and visibility degradation are estimated. These results improve our quantitative understanding of the development of haze events over the NCP and on formulating emission reduction strategies, as well as may also provide insights for other polluted regions.

## 2 Measurements and Methods

### 2.1 Location and instrumentation

Measurements were conducted within the framework of the BEST-ONE (Beijing winter finE particle STudy- Oxidation, Nucleation, and light Extinctions) field campaign from January 1 to March 5, 2016,



at the Huairou site (40.42°N, 116.69°E), located in a rural environment, north of Beijing, China. Detailed information about the sampling site was described in Tan et al. (2018). A weather station (Met one Instrument Inc., USA) was performed to measure meteorological parameters (ambient RH, temperature, wind speed, wind direction) and detailed aerosol particle physical and chemical properties were recorded using a suite of state-of-the-science instrumentation. Hygroscopic growth factor (HGF) of sub-micrometer aerosol particles was measured using a Hygroscopicity-Tandem Differential Mobility Analyzer (H-TDMA, TROPOS, Germany) (Wu et al., 2011;Massling et al., 2011;Wang et al., 2018;Wu et al., 2016;Liu et al., 1978) and data retrieval followed the TDMA$_{inv}$ method in Gysel et al. (2009). The hygroscopicity parameter ($\kappa$) was estimated using by the $\kappa$-Köhler approach (Petters and Kreidenweis, 2007;Köhler, 1936). Size-resolved NR-PM$_1$ was recorded by an Aerodyne High-Resolution Time-of-Flight Aerosol Mass Spectrometry (HR-ToF-AMS, Aerodyne Research, Inc., USA) (DeCarlo et al., 2006). Regular calibration procedures followed as reported in Jayne et al. (2000) and Jimenez et al. (2003) and composition dependent correction followed as in Middlebrook et al. (2012). Gaseous HNO$_3$ and NH$_3$ were measured using Gas-Aerosol Collector (GAC) coupled with Ion Chromatography (IC) (Dong et al., 2012). Mass concentration of equivalent black carbon in aerosol particles (Petzold et al., 2013) was recorded by Multi Angle Absorption Photometer (MAAP, Model 5012, Thermo Fisher Scientific, USA) with a laser wavelength of 670 nm (Petzold and Schönlinner, 2004). Furthermore, particle number size distribution (PNSD) in the size range of 3 nm~10 μm was measured using a Mobility Particle Size Spectrometer (MPSS, Model 3776+3085 3775+3081, TSI, USA), following the recommendations described in Wiedensohler et al. (2012) and an Aerodynamic Particle Size Spectrometer (APS, Model 3021, TSI, USA) (Wu et al., 2008;Pfeifer et al., 2016).

Detailed description on H-TDMA, HR-ToF-AMS and GAC-IC can be found in the supporting

information.

**2.2 Estimation of aerosol particle liquid water**

Given the absence of direct liquid water measurement, size-resolved liquid water was calculated using

the corresponding HGFs measured at RH=90% (50, 100, 150, 250, 350 nm in stokes diameter), PNSD

data (3 nm~10 μm) and meteorological parameters (RH, T), following the method proposed in Bian et

al. (2014), referred to below as H-TDMA-derived liquid water. Briefly, the measured PNSD with 57

size bins were fitted using a four-mode lognormal distribution. The classification of four modes and the

fitting results are shown in Table S1 and Figure S4. Good agreement between measured values and

fitted PNSD was achieved, which indicates the reliability of the four-mode lognormal fitting method.

Based on four-mode lognormal fitting results, the particle number size distribution and number fractions

of each mode can be obtained. It has been assumed that particles from the same mode have constant

particle hygroscopicity ($\kappa$). Under the assumption of constant particle hygroscopicity in each mode

(shown in Table S1), the $\kappa$ values for each mode ($\kappa_1$, $\kappa_2$, $\kappa_3$) can be calculated by equation [1] from the

known number fraction of fitted four modes and the $\kappa$ values of measured particle size from H-TDMA

measurement.

$$\kappa = \sum_{i=1}^{4} \kappa_i f_i \quad [1]$$

Here, $\kappa_i$ and $f_i$ represent the $\kappa$ value and the particle number fraction of the i mode. Then, the calculated

$\kappa$ values for each mode and the derived number fraction of each size bin were used to obtain the $\kappa$

distribution for each size bin. Figure S5 shows the comparison of calculated sized-resolved $\kappa$





distribution and the $\kappa$ measured by H-TDMA, the good agreement showed the reliability of the method.

Then, based on $\kappa$-Kölher theory (Petters and Kreidenweis, 2007;Köhler, 1936), the size-resolved *HGF*s

at ambient RH were calculated. Finally, liquid water of size-resolved particles can be derived by

calculating the differentials between the dry and wet PNSD of aerosol particles in equation [2]:

$$\text{Liquid water} = \frac{\pi}{6} N_j D_{p,j}^3 \left( HGF(D_p, RH)^3 - 1 \right) * \rho_w \qquad [2]$$

where j represents the bin number of measured PNSD, $N_j$ and $D_{p,j}$ represent the number concentration

and the diameter of dry particles of the $j^{th}$ bin, respectively, while, *HGF* and $\rho_w$, are the hygroscopic

growth factor of aerosol particles and water density (1 g/cm$^3$), respectively.

**2.3 Condensation rate of trace gases**

The condensation rate ($k$) of trace gases (dinitrogen pentoxide, $N_2O_5$ and nitric acid, $HNO_3$ in the

constrained conditions, referred as k_$N_2O_5$ and k_$HNO_3$ below) was calculated by the method of

Schwartz (1986), shown in equation [3]. In order to illustrate the influences of the dry and wet PNSD

due to water uptake on condensation rate of gases, the PNSD of the dry and wet particles (obtained by

applying the HGF estimated from H-TDMA-derived liquid water method) were used.

$$k = \frac{4\pi}{3} \int_0^\infty \left( \frac{r^2}{3D_g} + \frac{4r}{3C_g\gamma} \right)^{-1} r^3 \frac{dN}{dlogr} dlogr \qquad [3]$$

$$C_g = \sqrt{\frac{3RT}{M}} \qquad [4]$$

Where, $r$ represents radius of the particles, $D_g$ represents the binary diffusion coefficient evaluated

following Maitland (1981) (1.18*e$^{-5}$ m$^2$/s). $C_g$ is the kinetic velocity of the gas molecules, calculated in

equation [4]. Here, R and M are the ideal gas constant ($8.314 \ kg.m^2/mol/K/s^2$) and molar mass of the

gas, respectively while T represents the ambient temperature. dN/dlogr is the number size distribution

and $\gamma$ is the uptake coefficient of the gas.

The uptake coefficient of $N_2O_5$ was estimated following the method proposed in Chen et al. (2018b) and

Chang et al. (2016) and references therein. The uptake suppression effect of $N_2O_5$ due to the presence of

secondary organic aerosol (SOA) was considered following the method in Anttila et al. (2006). Based

on our source apportionment using Positive matrix factorization (SoFi tool, ME2, Francesco Canonaco,

PSI), two oxygenated organic aerosol factors (OOA), usually interpreted as SOA, and three primary

organic aerosol factors (POA) were determined. The fraction of SOA in the total organic aerosol (OA)

was 60%~90% during the observed period, which is quite consistent with the results of a previous study

in Beijing (Huang et al., 2014). Hence, 75% was used as the ratio of SOA/OA in our model to calculate

uptake coefficient of the $N_2O_5$, where the suppression effect of SOA on the uptake of $N_2O_5$ was

estimated following the work of Anttila et al. (2006). Additionally, the reaction of chloride with $N_2O_5$

was not considered in this study due to its limited mass concentration (on average 5% of the $PM_1$ mass

concentration during the marked haze period), which might cause uncertainty in the $k\_N_2O_5$ calculation.

The detailed information regarding the estimation $\gamma_{N2O5}$ is given in Chen et al. (2018b). For the

estimation of $\gamma_{HNO3}$, it was reported that the $\gamma_{HNO3}$ on the solid and deliquesced inorganic compound

such like sodium chloride were 0.01~0.03 (Fenter et al., 1994;Leu et al., 1995;Beichert and Finlayson-

Pitts, 1996) and >0.2 (even 0.5) (Guimbaud et al., 2002;Abbatt and Waschewsky, 1998), respectively.

Therefore, $\gamma_{HNO3}$=0.01 and $\gamma_{HNO3}$=0.5 are selected to calculate the lower and upper limit of condensation

rate of $HNO_3$ in the atmosphere.


## 2.4 Equilibrium of NH₄NO₃

The equilibrium dissociation constant of NH$_4$NO$_3$ ($Kp$) under dry conditions was calculated as a
function of ambient temperature (Seinfeld. and Pandis., 2006) in the following equation [5].

$$lnK_p = 84.6 - \frac{24220}{T} - 6.1ln\left(\frac{T}{298}\right) \quad [5]$$

Taking into account the associated liquid water, the equilibrium vapor pressure of HNO$_3$ was calculated
by employing the Extended-Aerosol Inorganic Model (E-AIM) Model II H$^+$ - NH$_4^+$ - SO$_4^{2-}$ - NO$_3^-$ - H$_2$O

(Clegg et al., 1998) using HR-ToF-AMS data, NH$_3$ from GAC-IC, and meteorological parameters (RH,
T). In this calculation, a simplified ion pairing scheme was performed to ensure the ion balance of the
input chemical composition following the method in Gysel et al. (2007).

## 2.5 Light extinction coefficient and visibility calculation

Size-resolved chemical composition of the NR-PM$_1$ from HR-ToF-AMS, mass concentration of

equivalent black carbon from MAAP, PNSD data and the H-TDMA-derived liquid water were used to
calculate light extinction coefficient (including light absorption and scattering) and visibility
degradation of size-resolved particles by the Mie scattering theory described in Barnard et al. (2010).
Here, size-resolved equivalent black carbon mass concentration was inferred by the particle mass size
distribution measurement from single particle soot photometer in PKUERS. The method of re-

distribution of liquid water and HR-ToF-AMS data has been described in the supporting information
(Text S1, HR-ToF-AMS introduction section). Thus, with the re-distributed datasets as the input of the
Mie scattering theory, the light extinction coefficient for atmospheric particles in the absence and

presence of liquid water with a size range of 100~2500 nm in stokes diameter can be derived. Due to

lack of measurements on aerosol particle morphology and mixing state, we assume particles are

spherical as described in Barnard et al. (2010). To perform Mie calculation, the complex reflective

index of each component is given in Table 1 of Barnard et al. (2010) and references therein. This

method shows good agreement with measurements in Mexico City and is consistent as the regional

atmospheric chemistry model WRF-Chem. Here, Ext_550nm_wet and Ext_550nm_dry represent the

calculated light extinction coefficient for particles in the presence and absence of liquid water at an

incident light wavelength of 550 nm. The corresponding visibility degradation (VIS) for dry/wet

particles was calculated from the light extinction coefficient following the Koschmieder equation [6].

$$VIS = \frac{3.912}{Ext\_5\ 5\ nm} \qquad [6]$$

## 3 Results and Discussion

### 3.1 Nitrate-dominated fine particulate matter pollution

Figure 1 illustrates a summary of chemical composition of NR-PM$_1$, ambient RH, size distribution and

total aerosol particle liquid water, size distribution and total aerosol surface area concentration during

the period of February 29 to March 5, 2016 in the BEST-ONE campaign. During this period, polluted

episodes occurred under stagnant meteorological conditions with low wind speed (Figure S6) and

elevated ambient RH (Figure 1a). As marked 'haze period' in Figure 1, an obvious increase of NR-PM$_1$

was observed. The secondary inorganic components (sulfate, nitrate and ammonium) were dominant

components of the NR-PM$_1$, accounting for up to 73% during the 'haze period'. Particularly, nitrate was

the major contributor of the secondary inorganic components and accounted for up to ~44% of NR-PM$_1$ mass, while sulfate contributed for ~12% on average.

In the recent decade, severe haze events with high aerosol mass loading occurred frequently in Beijing during wintertime (Hu et al., 2016;Hu et al., 2017;Sun et al., 2014;Sun et al., 2015). To mitigate the air pollution, the Beijing government implemented strict emission controls. The total mass loading of particulate matter has reduced substantially in the recent years (http://sthjj.beijing.gov.cn/). With decreasing in PM mass concentration, the mass fraction of particulate nitrate during these haze events in Beijing enhanced substantially. In 2014, the highest fraction of nitrate in PM$_1$ was reported as ~20% and increased to ~35% in 2016 (Xu et al., 2019b), which is comparable to the ratio (44%) in this study. The particulate nitrate became more dominant in secondary inorganic compounds other than particulate sulfate with the air quality improvement over NCP.

As one of the main hydrophilic compounds in atmospheric aerosol particles, the ability of water uptake at 90% RH of particulate NH$_4$NO$_3$ is comparable with particulate (NH$_4$)$_2$SO$_4$ (Kreidenweis and Asa-Awuku, 2014;Wu et al., 2016). However, compared to (NH$_4$)$_2$SO$_4$, NH$_4$NO$_3$ particles have a lower deliquescence RH (62%, 298 K) than (NH$_4$)$_2$SO$_4$ (80%, 298 K) (Kreidenweis and Asa-Awuku, 2014), and easily liquify (Li et al., 2017b). In addition, NH$_4$NO$_3$ particles are semi-volatile, the co-condensation of semi-volatile compounds and water (Topping et al., 2013;Hu et al., 2018) could be significant. Therefore, the switching from sulfate-dominated to nitrate-dominated aerosol chemistry may impact on aerosol water uptake. The interaction between aerosol particle liquid water and particulate nitrate formation and visibility degradation should be reconsidered.



## 3.2 Mutual promotion effects between liquid water and nitrate formation

In the following discussion, the high fraction of particulate nitrate during the 'haze period' is elucidated by theoretical calculations considering the uptake of $N_2O_5$ and $HNO_3$, and the thermodynamic

equilibrium of $NH_4NO_3$. In particular, the role of aerosol water uptake in particulate nitrate formation is comprehensively investigated.

 $N_2O_5$ is an important gaseous precursor for nitrate formation via its hydrolysis to form $HNO_3$ during nighttime (Brown et al., 2006). Liquid water can enhance aerosol surface areas and volumes, thereby increasing the available heterogeneous reacting medium. Across the development of 'haze period', the

estimated liquid water increased from ~1 µg/m$^3$ at the beginning ($2^{th}$ March, 14:00~18:00 p.m.) to ~75 µg/m$^3$ when the haze was fully developed ($4^{th}$ March, 4:00~8:00 a.m.). The total surface area and volume concentrations of particles were increased by the liquid water by 2~3% at the beginning and by about 25~40% in the fully-developed haze compared to the 'dry' values, respectively (see Figure S7 and S8). Additionally, from the beginning to the fully-developed haze, the uptake coefficient of $N_2O_5$ was

enhanced by a factor of 9 from 0.002 to 0.018, and the k_$N_2O_5$ increased by a factor of 20 (dry particles); while, considering the increased particle surface area and volume due to water uptake, the respective value of enhanced k_$N_2O_5$ was 25 (Figure 2a). Apart from providing extra reacting medium, the abundant liquid water can liquefy the aerosol particles and may reduce any kinetic limitation of mass transfer for reactive gases (Koop et al., 2011;Shiraiwa et al., 2011) and impact thermodynamic

equilibrium of semi-volatile compounds (Kulmala et al., 1993;Topping et al., 2013) to contribute to secondary aerosol formation. Our previous study provided the observational evidence that particles may have transitioned from the solid phase to the liquid phase as RH increased from 20% to 60% during

wintertime in Beijing (Liu et al., 2017). In this study, the ambient RH increased from ~10% up to 70%

during the haze period, suggesting a likely transition of particles from the solid to liquid phase. Such

phase transition may facilitate particulate nitrate formation by increasing diffusion coefficients of

dissolved precursors.

To illustrate the facilitation of nitrate formation in the presence of liquid water, we performed the

theoretical calculation of equilibrium between particulate $NH_4NO_3$ and gaseous $HNO_3$ under dry and

ambient conditions, respectively. First, the dissociation constant of $NH_4NO_3$ ($Kp$) was calculated using

equation [5] without considering the influence of the liquid water. $Kp$ ranged from 0.06 (275.3 K) to

4.61 (291.5 K) $ppb^2$ during the 'haze period'. The measured partial pressure product (2.55~9.63 $ppb^2$)

was greater than the equilibrium $Kp$ nearly all the time (Figure 3). In this case, gaseous $NH_3$ and $HNO_3$

in the atmosphere were supersaturated and would tend to partition into the dry particle phase gradually

even in the absence of liquid water. The presence of liquid water under ambient RH can supress the

$HNO_3$ equilibrium vapor pressure to nearly zero, changing equilibrium and facilitate the partitioning of

nitrate substantially. The equilibrium vapor pressure of $HNO_3$ over particles was calculated by E-AIM

Model II (www.aim.env.uea.ac.uk) taken into account the liquid water. Note that this calculation

assumes negligible interaction between dissolved organic components and the activity of $NO_3^-$. In the

presence of aerosol associated water, the $HNO_3$ equilibrium vapor pressure dropped from its dry values

to effectively zero, indicating liquid water significantly favored greater partitioning to particulate

nitrate. The negligible equilibrium vapor pressure of $HNO_3$ resulted in essentially no $HNO_3$ evaporation

back to the gas phase and irreversible uptake of $HNO_3$ can be assumed under the ambient RH and $NH_3$

concentration. This enabled the simplified treatment of the irreversible condensation rate following

Schwartz (1986) used below. As shown in Figure 4, the partitioning ratio (molar ratio between particulate and total nitrate) increasing with RH was observed during the development of haze, and 98% of nitrate was present as particle phase when the haze was fully developed with liquid water increasing from 1 µg/m$^3$ to ~75 µg/m$^3$.

Furthermore, the presence of aerosol associated water was substantially enhanced by the uptake rate of HNO$_3$, which could dominate the gaseous HNO$_3$ partitioning into particle phase throughout the haze developing. Because the negligible equilibrium vapor pressure suggests that HNO$_3$ condensation loss was not limited by thermodynamic equilibrium but limited by its uptake rate. The condensation (or uptake) rate of HNO$_3$ (k_HNO$_3$) can be calculated using equations [3-4]. Here, the lower and upper limit of k_HNO$_3$ were calculated assuming the uptake coefficient (γ) of HNO$_3$ in the range of 0.01 to 0.5 (Fenter et al., 1994;Leu et al., 1995;Beichert and Finlayson-Pitts, 1996;Abbatt and Waschewsky, 1998;Guimbaud et al., 2002). As shown in Figure 2b and 2c, the lower (upper) limit of k_HNO$_3$ increased by a factor of 2.9 (2.7) for dry PNSD and 3.5 (3.1) for wet PNSD from the beginning to fully-developed haze period. As one can see, the liquid water facilitated the rate of HNO$_3$ uptake and hence the particulate nitrate formation.

The above analyses quantify the effect of the increased aerosol surface area and volume concentrations resulting from the water uptake on the particulate nitrate formation through increased uptake of N$_2$O$_5$ and HNO$_3$. Such an effect becomes more pronounced with the increasing pollution throughout the haze event owing to the simultaneously increasing ambient RH. Owing to its hygroscopicity, the increased ammonium nitrate mass fraction led to a further increase in aerosol surface area and volume

concentrations through additional increase in liquid water, further enhancing uptake of condensable

vapors.

It is worth noting that a similar co-condensation effect between water vapor and semi-volatile organic components (Topping and McFiggans, 2012;Topping et al., 2013;Hu et al., 2018) could promote the haze formation as well, for which there may be some evidence in the current case. Such a co-condensation effect will lead to the enhancement of semi-volatile organic and inorganic (e.g., nitrate)

material with the increasing RH in a developing haze. The associated water will favor partitioning of both particulate nitrate and semi-volatile organic materials to the particle phase depending on the organic solubility, providing a linkage between the development of increasing organic and inorganic particle mass.

### 3.3 The key role of liquid water on visibility degradation

Aerosol particles grow up in size as ambient RH increases, further enhances their extinction coefficient and impacts visibility (Zhao et al., 2019;Kuang et al., 2016). In this section, size-resolved extinction coefficient of aerosol particles was estimated, and the influences of liquid water on the extinction coefficient and visibility were quantitatively evaluated. As shown in Figure 5a, the total light extinction coefficient of dry and wet aerosol particles enhanced by a factor of 4.3 and 5.4, respectively, from the

beginning to a fully-developed haze. Correspondingly, the calculated visibility without considering liquid water degraded significantly from ~10 km to less than 2 km within 48 hours during the marked 'haze period'. The contribution of aerosol associated water to visibility impairment was negligible in the

beginning (2%), while it was significant (up to 24%) in the fully-developed haze (Figure 5b). This indicates that liquid water facilitated visibility degradation during haze development.

The influences of liquid water on visibility degradation varied with aerosol particle size. The size-resolved chemical composition data showed that the inorganic species, mainly nitrate, were dominant components in the aerosol particles within the size range of 300~700 nm (Figure S3). Correspondingly, the particles in this size range contained most of the liquid water (50~80% of the total aerosol liquid water content of $PM_1$). According to discussion in Sec. 3.2, the mutual promotion effect between liquid

water and particulate nitrate can promote their formation. Aerosol particles in this size range experienced the most significant enhancement of light extinction due to water uptake (Figure 6a and 6b) and contributed 70~88% of the total extinction coefficient of the total $NR-PM_1$ (Figure S9). In conclude, the rapid nitrate formation enhanced the aerosol extinction coefficient during haze developing, while the aerosol water uptake further enhanced the visibility degradation by increasing

extinction coefficient and promoting nitrate formation.

It is worth noting that the enhanced dimming effect will further shallower the planetary boundary layer (PBL), which, in turn, depresses the dilution of water vapor and particulate matter in the atmosphere, hence leads to a higher RH and aerosol particle mass loading (Tie et al., 2017). Such effect is beyond the scope of this study.



## 4 Conclusions and implication

In this study, we observed a nitrate-dominated (up to 44% of non-refractory PM$_1$ mass concentration) particulate matter pollution episode, which is typical during winter haze in Beijing, China. A clear co-increase of aerosol particle liquid water and particulate nitrate was observed, demonstrating the mutual promotion effect between them via observation-based theoretical calculations.

As shown in Figure 7, the water uptake by hygroscopic aerosols increased the aerosol surface area and volume, favoring the thermodynamic equilibrium of ammonium nitrate and enhancing the condensational loss of N$_2$O$_5$ and HNO$_3$ over particles. The enhanced particulate nitrate formation from the above pathways increased the mass fraction of particulate nitrate, which had a lower deliquescence RH than sulfate and resulted in more water uptake at lower ambient RH (Kreidenweis and Asa-Awuku, 2014). Hence, the increased aerosol particle surface area and volume concentrations due to water uptake, in turn facilitates particulate nitrate formation. Hence, a feedback loop between liquid water and particulate nitrate is built up. Therefore the enhanced particulate nitrate components can accelerate the feedback compared with sulfate-rich pollution over the NCP region in the past (Hu et al., 2016). This self-amplification can rapidly degrade air quality and halve visibility within one day. Our results highlight the importance of reducing the particulate nitrate and its precursors (e.g. NO$_x$) for mitigation of haze episodes in NCP region.

## Data availability

The observational dataset of the BEST-ONE campaign can be accessed through the corresponding author Z. Wu ([zhijunwu@pku.edu.cn](mailto:zhijunwu@pku.edu.cn)).

The E-AIM model can be accessed via [http://www.aim.env.uea.ac.uk/aim/aim.php](http://www.aim.env.uea.ac.uk/aim/aim.php).

## Author contributions

Z.W., Y.W. and Y.C. conceived the study. Y.Z., M.H., and A.K.S developed BEST-ONE field campaign program. Y.W., Z.W., D.S., Z.D., S.H.S., R.S., G.I.G., P.S., T.H., K.L., L.Z., C.Z., A.K.S., Y.Z., and M.H. participated in this campaign and collected the dataset. Y.W. conducted aerosol particle

liquid water calculation under guide of Y.B. and thermodynamic equilibrium of particulate ammonium nitrate under guidance of G.M. Y.C. calculated the uptake coefficient of $N_2O_5$, optical properties and visibility. Y.W. and Y.C. cowrite the manuscript with the inputs from all co-authors. Z.W., G.M., A.K.S., S.H.S., G.I.G., P.S., T.H., A.V., and A.W. proofread and help improve the manuscript. All authors discussed the results.

## Acknowledgement

This work is supported by the following projects: National Natural Science Foundation of China (41571130021, 41875149), Ministry of Science and Technology of the People's Republic of China (2016YFC0202801), German Federal Ministry of Education and Research (ID-CLAR). Y.W. acknowledges the support of the China Scholarship Council and The University of Manchester Joint

Scholarship Programme. We thank Dr. Paul I. Williams for valuable advice on reaction constant of $HNO_3$ and $N_2O_5$.



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








**Figure 1: The time series of (a) NR-PM₁ chemical composition measured by the HR-ToF-AMS and ambient RH (red solid line), (b) size-segregated aerosol particle liquid water and the total mass concentration of liquid water with smaller than 1 μm in aerodynamic diameter (red solid line), (c) size-segregated aerosol particle surface area and total aerosol particle surface area**

**without considering particle hygroscopic growth.**





**Figure 2: The time series of (a) condensation rate of N₂O₅ (k_N₂O₅) with the calculation of dry particle number size distribution (PNSD) and wet PNSD, (b-c) condensation rate of HNO₃ (k_HNO₃) with the calculation of dry and wet PNSD under the assumption of γ=0.01 and γ=0.5, respectively during February 29 to March 5, 2016.**





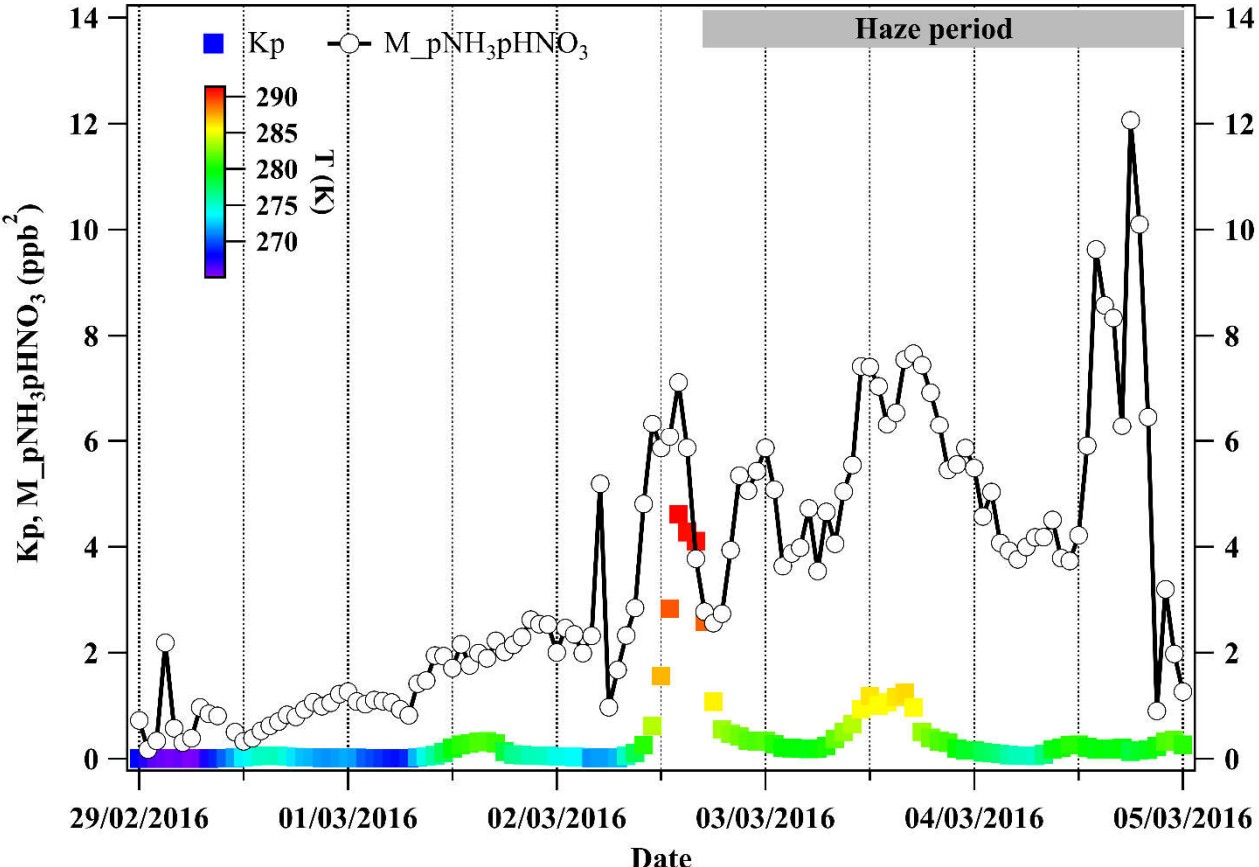

**Figure 3: The comparison of the calculated temperature-dependent dissociation constant of NH₄NO₃ (Kp) (Seinfeld. and Pandis., 2006) in the absence of liquid water and the product of mixing ratios of gaseous NH₃ and HNO₃ measured by GAC-IC (M_pNH₃pHNO₃). Here, Kp is colored by the ambient temperature ranging 265~293K during February 29 to March 5, 2016.**



**Figure 4: The relationship between aerosol particle liquid water and $m_{NH_4NO_3}/(m_{HNO_3} +$**

$m_{NH_4NO_3})$ **(left axis) and mass concentration of NH₄NO₃ in the particle phase (right axis) during**

**the period of February 29 to March 5, 2016. Here, NH₄NO₃ in the particle phase was measured by**

**HR-ToF-AMS and the HNO₃ in the gas phase was measured by GAC-IC. Liquid water was**

**calculated by H-TDMA-derived method.**



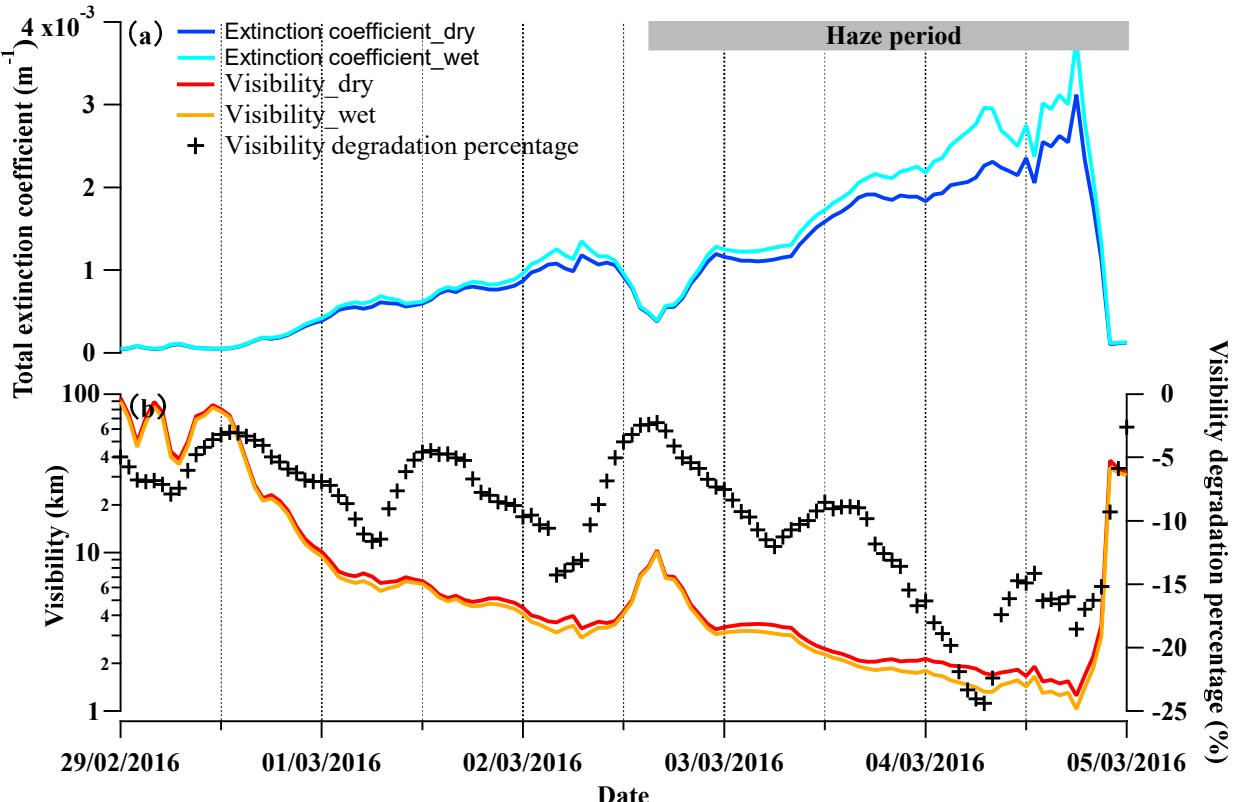

**Figure 5: The time series of (a) calculated total extinction coefficient at wavelength of 550 nm with the consideration of dry and wet PNSD, referred as Extinction coefficient_dry and Extinction coefficient_wet, (b) calculated visibility with the consideration of dry and wet PNSD, referred as Visibility_dry and Visibility_wet, respectively. Visibility degradation percentage is (Visibility_wet-Visibility_dry)/Visibility_dry, representing the visibility degradation in the presence of liquid water.**



**Figure 6: (a) Size-segregated light extinction coefficient at wavelength of 550 nm for wet particles (Extinction coefficient_wet), (b) size-segregated difference between Extinction coefficient_wet and Extinction coefficient_dry, representing light extinction coefficient difference with and without considering liquid water.**





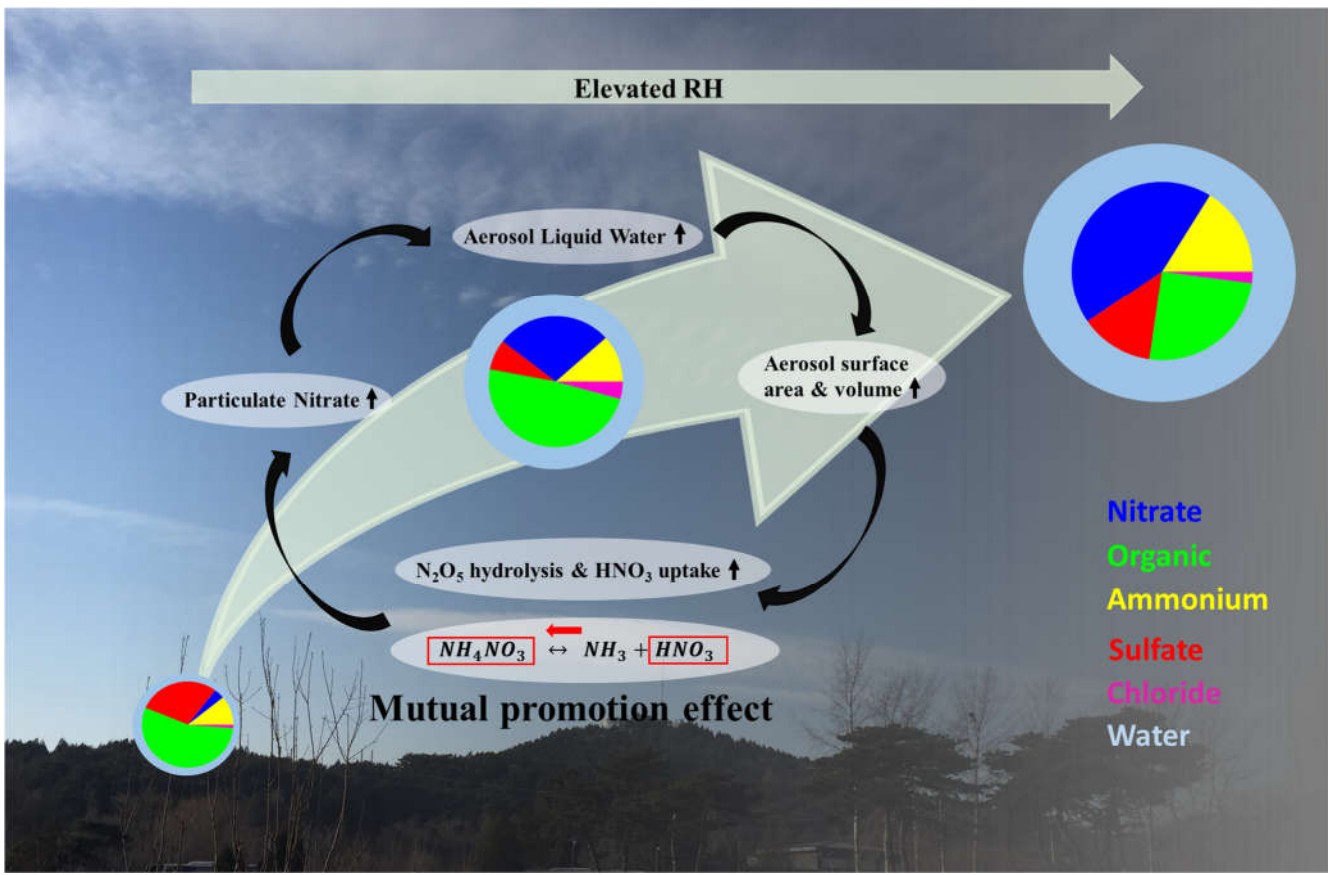

**Figure 7: The scheme of the mutual promotion effect between aerosol liquid water and particulate nitrate**

