# Peer review of "Supporting Information for"

_Atmospheric Chemistry and Physics, 2019_

## Short Comment (SC1) · 1 Oct 2019

The subject of the paper is the uptake of dinitrogen pentoxide onto PM including the hydrolytic uptake on aqueous aerosol. The process is very likely to occur in GAC instruments and the resulting nitrate measured by the IC will very likely contain a substantial contribution from the hydrolysis of nitrogen pentoxide within the GAC sampling system, where two nitrate ion result from the sampling of one dinitrogen pentoxide molecule. In situations where the chemistry of the N2O5/nitric acid/PM system is being studied, I would think that a consideration of this issue would be important at the very least in

estimating the uncertainties of the experimental approach.

A description of the effect as observed in the Applikon MARGA was published in Phillips, G. J., Makkonen, U., Schuster, G., Sobanski, N., Hakola, H., and Crowley, J. N.: The detection of nocturnal N2O5 as HNO3 by alkali- and aqueous-denuder techniques, Atmos. Meas. Tech., 6, 231–237, https://doi.org/10.5194/amt-6-231-2013, 2013. All instruments using the same or similar means of determining nitric acid will very likely suffer form this effect.

Have the authors considered this effect? What uncertainty does it introduce into the data analysis?

---

## Referee Comment (RC1) · Anonymous Referee #1 · 24 Oct 2019

Wang et al. reported their field observation of chemical composition in particulate matter (PM) in Beijing. Based on the data together with measured meteorological parameters, they estimated condensational loss rate of nitrate precursor gases for nitrate formation and aerosol liquid water. In addition, they discussed the interplay between increased particulate nitrate and aerosol particle liquid water, suggesting that particulate nitrate increased the aerosol liquid water and the aerosol liquid water promoted condensational loss of nitrate precursor gases ($N_2O_5$ and $HNO_3$), i.e., increasing particulate nitrate. This study has highlighted an increasing importance of particulate nitrate during a polluted episode in wintertime in Beijing, and would be an important contribution to the understanding of recent change in chemical composition of PM. I would recommend this manuscript for publication in ACP after addressing the following issues.

Specific comments:

Lines 85-86: The statement on "via well chemical pathways" is a bit misleading for the oxidation of SO2 because sulfate production mechanisms from the SO2 oxidation during haze episodes in Beijing are still ongoing debate (Cheng et al., 2016; Gen et al., 2019).

Cheng et al., Science Advances 2.12 (2016): e1601530.
Gen et al., Environmental Science & technology, (2019): 53, 8757-8766.

I can understand that nitrate mass fraction in secondary inorganic components has increased. However, I am not sure why the absolute mass concentration of nitrate can increase despite the reduction of NOx over North China. More discussion on this point deserves to appear in the introduction.

The authors need to explain how they estimate the uptake coefficient of N2O5 because they discussed factors affecting the coefficients.

Lines 268-270: I think that this depends on RH. Need to verify it.

Lines 312-313: It can be understood that in their calculation the authors ignore the interaction between dissolved organic components and the activity of NO3-. However, small organic acids affect the aerosol hygroscopicity. Can the authors provide the potential impact of such small organic acids on their calculation?

Molar fraction of nitrate over total of HNO3 + NO3- highly depends on particle pH (e.g., Nah et al., 2018). The authors attributed a strong positive correlation between particulate nitrate and aerosol liquid water to the feedback loop between them, but no particle pH effect has been discussed. Can you exclude the possibility that increased particle pH would lead to an increase in particulate nitrate?

Nah et al., Atmospheric Chemistry and Physics 18.15 (2018): 11471-11491.

If the mutual effect of both nitrate and aerosol particle liquid water is the determine factor for increasing nitrate, why did the mass concentration of ammonium nitrate reach a plateau at aerosol liquid water of > 20 µg/m3?

Finally, an increase in RH also leads to increasing aerosol liquid water. As shown in Fig. 1, the increased nitrate is coincident with the elevated RH during the haze period. How do the authors discriminate the mutual effect from this elevated RH effect?

Minor comments:
Lines 109-113: Is the discussion about NO3-/SO42- in China?
Line 123: Need to be more explicit about how the reactive uptake o precursors and the thermodynamic equilibrium of ammonium nitrate change. Enhancing or increasing?
Line 279: theoretical calculations of what?
Line 293: Please use either liquify or liquefy for consistency.
Line 346: HNO3 instead of particulate nitrate?

---

## Referee Comment (RC2) · Anonymous Referee #2 · 22 Nov 2019

This study presented observation results and theoretical calculations, and thereby proposed the so-called "mutual promotion effect between aerosol particle liquid water and nitrate formation". This study focused on an interesting topic, and is presented in a relatively clear way. However, the major problem I see is the confusion between "equilibrium" and "formation". I would recommend this manuscript for publication in ACP only if the following concerns can be nicely addressed.

As described above, this study kind of mixed up the concept of "equilibrium" and "formation". Now let's assume a system without N2O5. If the aerosols are already del-

iquencent, as the RH increase, the liquid water content (LWC) will increase, and the gas-particle partitioning of both NH3 and HNO3 will be influenced. Indeed, the final result might be that more HNO3 partitioned on the pariticle phase and increasing the LWC, until a new equilibrium is reached that with more LWC and higher particle-phase NO3- fraction. However, this process should be viewed as a new "equilibrium" driven by the elevated RH, and not "formation" of HNO3. In this sense, there's no so-called "mutual promotion formation" concept. A lot of factors could influence this equilibrim, including aerosol acidity, activity and phase partitioning, but in any sense, there's no "formation" in terms of total HNO3 in the system.

Moreover, the authors seems to argue that that although gas-phase NH3 and HNO3 are also "supersaturated" under dry conditioned (in comparison with the equilibrium dissociation constant of NH4NO3 (Kp) under dry conditions), their condensation loss is "limited by its uptake rate". Once the aerosols get deliquescent, the update rate would be accelerated and more NO3- will be present onto the particle phase. I agree this process may happen during the initial dry-wet transition periods, but what's the timescale? I would say that the system would transfer from the "rate-limited regime" under dry conditions to "equilibrium-controlled regime" soon after this transition, on the scale of hours. Afterwards, as I described above, the apparent increase in NO3 is only a result of equilibrium moving due to RH variations, if there're no influences from other factors. In this sense, I don't get the where the "mutual promotion" lies in – if it's referring merely to those several hours, I won't consider it as an important process in the haze development.

The only way that I would agree the concept of "mutural promotion formation" is that, the uptake by N2O5 under high RH is so important that it contributed signicantly to the total HNO3. This is not shown in this study. I would suggest to distinguish the relative contribution of HNO3 partitioning and N2O5 uptake on the final enhancement of particle-phase NO3-.

---

## Author Comment (AC1) · 2 Jan 2020

**Response to Gavin Philips:**

The subject of the paper is the uptake of dinitrogen pentoxide onto PM including the hydrolytic uptake on aqueous aerosol. The process is very likely to occur in GAC instruments and the resulting nitrate measured by the IC will very likely contain a substantial contribution from the hydrolysis of nitrogen pentoxide within the GAC sampling system, where two nitrate ion result from the sampling of one dinitrogen pentoxide molecule. In situations where the chemistry of the N2O5/nitric acid/PM system is being studied, I would think that a consideration of this issue would be important at the very least in estimating the uncertainties of the experimental approach.

A description of the effect as observed in the Applikon MARGA was published in Phillips, G. J., Makkonen, U., Schuster, G., Sobanski, N., Hakola, H., and Crowley, J. N.: The detection of nocturnal N2O5 as HNO3 by alkali- and aqueous-denuder techniques, Atmos. Meas. Tech., 6, 231–237, https://doi.org/10.5194/amt-6-231-2013, 2013. All instruments using the same or similar means of determining nitric acid will very likely suffer from this effect.

Have the authors considered this effect? What uncertainty does it introduce into the data analysis?

*Thanks for your valuable comments about this interference of $N_2O_5$ on $HNO_3$ detection of GAC-IC. We have tested the potential interference of $N_2O_5$ in the studied period, and added the corresponding discussion into the revised manuscript.*

*To test the potential $N_2O_5$ interference within investigated period, we re-grouped the measured dataset into the daytime (7:00 ~ 18:00 LT) and nighttime (18:00 ~ 07:00$^{+1}$ LT) periods. As the rapid photolysis of $NO_3$ radical and limited $N_2O_5$ concentration during daytime, $N_2O_5$ and its interference on GAC measurement are negligible during daytime. As shown in the revised Figure 4 below, we found a consistent pattern of molar ratio of particulate nitrate in the total nitrate ($m_{NH_4NO_3}/(m_{NH_4NO_3} + m_{HNO_3})$) as a*

*function of aerosol liquid water during the nighttime (green solid triangle) and daytime periods (red solid triangle). During the nighttime, the interference of $N_2O_5$ contributes higher $HNO_3$ in GAC measurement. That means, an underestimation of the $m_{NH_4NO_3}/(m_{NH_4NO_3} + m_{HNO_3})$ during nighttime was expected. This is consistent with the slightly underestimation (about 4%~8%) of the $m_{NH_4NO_3}/(m_{NH_4NO_3} + m_{HNO_3})$ during the nighttime when aerosol particle liquid water is less than 10 μg/m³, as shown in revised Figure 4.*

*As discussed above, the interference of $N_2O_5$ on $HNO_3$ observations is not expected to change the conclusions of our study during the investigated period in Beijing. We have added the discussion of this $N_2O_5$ interference in the revised manuscript (line 312-317, revised clean version manuscript) as shown below:*

*"The function between the particulate nitrate fraction and aerosol liquid water is given in Figure 4. It is worth noting that $N_2O_5$ hydrolysis during nighttime can contribute extra $HNO_3$ in the wet denuding method within GAC-IC system. This effect might explain the slightly underestimation of the particulate nitrate fraction in the total nitrate during nighttime when aerosol liquid water is less than 10 μg/m³ (Figure 4). However, the general consistency of this function between daytime and the nighttime (Figure 4) suggests a negligible influence of $N_2O_5$ interference on our analysis during the investigated period."*

[Figure]

*Figure 4: The relationship between aerosol particle liquid water and the molar ratio of particulate nitrate in the total nitrate, $m_{NH_4NO_3}/(m_{HNO_3} + m_{NH_4NO_3})$ (left axis) during the nighttime 18:00~07:00+1 (green solid triangle) and the daytime at 07:00 ~ 18:00 (red solid triangle), and mass concentration of particulate nitrate as a function of aerosol liquid water (right axis) during the period of during February 29 to March 5, 2016. Here, particulate nitrate was measured by HR-ToF-AMS and the HNO₃ in the gas phase was measured by GAC-IC. Aerosol liquid water was calculated by H-TDMA-derived method.*

---

## Author Comment (AC2) · 2 Jan 2020

**Response to comments of anonymous referees # 1**

**General comments**

Wang et al. reported their field observation of chemical composition in particulate matter (PM) in Beijing. Based on the data together with measured meteorological parameters, they estimated condensational loss rate of nitrate precursor gases for nitrate formation and aerosol liquid water. In addition, they discussed the interplay between increased particulate nitrate and aerosol particle liquid water, suggesting that particulate nitrate increased the aerosol liquid water and the aerosol liquid water promoted condensational loss of nitrate precursor gases ($N_2O_5$ and $HNO_3$), i.e., increasing particulate nitrate. This study has highlighted an increasing importance of particulate nitrate during a polluted episode in wintertime in Beijing, and would be an important contribution to the understanding of recent change in chemical composition of PM. I would recommend this manuscript for publication in ACP after addressing the following issues.

*Many thanks to the reviewer for the comments and suggestions. We have improved the manuscript accordingly. Please find a point-by-point response below.*

**Specific comments:**

**Q1.** Lines 85-86: The statement on "via well chemical pathways" is a bit misleading for the oxidation of $SO_2$ because sulfate production mechanisms from the $SO_2$ oxidation during haze episodes in Beijing are still ongoing debate (Cheng et al., 2016; Gen et al., 2019).

Cheng et al., Science Advances 2.12 (2016): e1601530.

Gen et al., Environmental Science & technology, (2019): 53, 8757-8766.

*Thanks for the useful suggestion. The reviewer is right that the particulate sulphate formation is still ongoing debate, not well-established yet.*

*The original sentence has been revised as shown below.*

*"In atmospheric environments influenced by anthropogenic activities, particulate secondary inorganic compounds are often dominated by particulate sulfate and nitrate (Heintzenberg, 1989), which originate from the oxidation of sulfur dioxide ($SO_2$) and nitrogen oxides ($NO_x$) via multiple chemical pathways (Calvert et al., 1985;Cheng et al., 2016;Wang et al., 2016;Gen et al., 2019a, b)."*

**Q2.** I can understand that nitrate mass fraction in secondary inorganic components has increased. However, I am not sure why the absolute mass concentration of nitrate can increase despite the reduction of $NO_x$ over North China. More discussion on this point deserves to appear in the introduction.

*Thanks for your helpful comment. The description of increasing nitrate mass concentration by 7% in the manuscript between 2006 and 2015 was from model simulation with the input for $NO_x$ emissions of 20.8 Tg (2006) and assumed value of 24.26 Tg (2015) (Wang et al., 2013). However, $NO_x$ emissions showed a declining trend during the period of 2010~2017 (Liu et al., 2017;Vu et al., 2019;de Foy et al., 2016). The mass concentration of particulate nitrate showed a declining trend as well as sulfate during 2013~2017 (Wang et al., 2019).*

*The discussion on nitrate mass concentration trend was added in the Introduction and the corresponding context is rephrased, as shown below.*

*"In the recent years, China has also managed to reduce $SO_2$ emissions by 75% during 2007~2015 (Li et al., 2017) and declined by ~15.1% per year during 2013~2017 (Vu et al., 2019), whereas $NO_x$ emissions declined only by ~10% between 2011 and 2015 (de*

*Foy et al., 2016) and by ~ 4.3% per year during 2013~2017 (Vu et al., 2019). The strict emission control reduced the $PM_{2.5}$ mass concentration and the corresponding chemical components in China significantly (Vu et al., 2019). The annual mean $PM_{2.5}$ mass loading decreased by 39.6% during 2013~2017 in Beijing-Tianjin-Hebei region, and the $SO_4^{2-}$ and $NO_3^-$ mass concentrations in the $PM_{2.5}$ declined by 40% and 34% respectively during 2015~2017 in Beijing (Vu et al., 2019). However, $NH_3$ emissions have been observed by satellites to increase by ~30% from 2008 to 2016 over the North China Plain (NCP) (Liu et al., 2018). The faster reduction rate of $SO_2$ than $NO_x$ emissions in conjunction with elevated $NH_3$ level, made it reasonable of switching dominant inorganic component in fine aerosol particles from sulfate to nitrate in the recent years similar like European countries (Sun et al., 2015;Hu et al., 2017;Hu et al., 2016;Wu et al., 2018;Guo et al., 2014;Huang et al., 2014;Huang et al., 2010;Ge et al., 2017;Xu et al., 2019;Xie et al., 2019;Li et al., 2018). Field measurements in Beijing show that annually averaged $NO_3^-/SO_4^{2-}$ molar ratio of NR-PM$_1$ (non-refractory $PM_1$) in 2012 (1.3~1.8) (Sun et al., 2015) has significantly increased compared to that in 2008 (0.9~1.5) (Zhang et al., 2013). Comparably, the $NO_3^-/SO_4^{2-}$ molar ratio of $PM_{2.5}$ in Beijing increased substantially, from 1.5 before 2013 to 3.33 in 2017 (Xu et al., 2019)."*

**Q3.** The authors need to explain how they estimate the uptake coefficient of $N_2O_5$ because they discussed factors affecting the coefficients.

*We had briefly described how to estimate the uptake coefficient of $N_2O_5$ in the Method section. In the revised version, we have improved this description, provided a clearer summary of the method and refer more details to the corresponding references, as shown below.*

*"The uptake coefficient of $N_2O_5$ was estimated following the method proposed in Chen et al. (2018) and Chang et al. (2016) and references therein. The influences of RH, temperature, multiple inorganic particle compositions, secondary organic aerosol (SOA) and primary organic aerosol are considered. The uptake suppression effect of*

*N2O5 due to the presence of SOA was considered following the method in Anttila et al. (2006). Based on our source apportionment using Positive matrix factorization (SoFi tool, ME2, Francesco Canonaco, PSI), two oxygenated organic aerosol factors (OOA), usually interpreted as SOA, and three primary organic aerosol factors (POA) were determined. The fraction of SOA in the total organic aerosol (OA) was 60%~90% during the observed period, which is quite consistent with the results of a previous study in Beijing (Huang et al., 2014). Hence, 75% was used as the ratio of SOA/OA in our model to estimate the suppression effect of SOA on the uptake of $N_2O_5$ following the work of Anttila et al. (2006). The reaction of chloride with $N_2O_5$ was not considered in this study due to its limited mass concentration (on average 5% of the $PM_1$ mass concentration during the marked haze period), which could cause minor uncertainty in the $k\_N_2O_5$ calculation. The detailed information regarding the estimation $\gamma_{N2O5}$ is given in Chen et al. (2018), and influence of different chemical components on $\gamma_{N2O5}$ is summarized in the Table 1 of Chen et al. (2018)."*

**Q4.** Lines 268-270: I think that this depends on RH. Need to verify it.

*Good point. To verify this, I used the online thermodynamic model UManSysProp (http://umansysprop.seaes.manchester.ac.uk/) to calculate the hygroscopic growth factor (GF) of pure $NH_4NO_3$ and $(NH_4)_2SO_4$ particles as a function of RH above their deliquescence RH. As shown in Fig. R1, for the deliquescent pure $NH_4NO_3$ and $(NH_4)_2SO_4$ particles, the hygroscopic growth factor under the same RH was comparable.*

[Figure]

*Fig. R1. Calculated hygroscopic growth factor (GF) at 298 K with 100 nm dry particle size for pure NH$_4$NO$_3$ and (NH$_4$)$_2$SO$_4$ as a function of relative humidity (RH) from UManSysProp (*http://umansysprop.seaes.manchester.ac.uk/index*). Here, the co-condensation of water and NH$_4$NO$_3$ was not considered in the calculation.*

*The original sentence has been revised as shown below.*

*As one of the main hydrophilic compounds in atmospheric aerosol particles, the ability of water uptake is comparable between deliquescent (NH$_4$)$_2$SO$_4$ and NH$_4$NO$_3$ particles with same sizes and ambient RH (Kreidenweis and Asa-Awuku, 2014;Wu et al., 2016), (http://umansysprop.seaes.manchester.ac.uk/).*

**Q5.** Lines 312-313: It can be understood that in their calculation the authors ignore the interaction between dissolved organic components and the activity of NO$_3^-$. However, small organic acids affect the aerosol hygroscopicity. Can the authors provide the potential impact of such small organic acids on their calculation?

*In this study, the aerosol liquid water was calculated based on the measurements of the particle number size distribution, hygroscopicity by H-TDMA, ambient RH and T. Here, the hygroscopicity measurement gives an bulk hygroscopic growth of given particle size. Therefore, the estimated aerosol liquid water has already considered the contribution*

*from organic compounds. Then, the aerosol liquid water was used as input of E-AIM model to calculate the equilibrium vapor pressure of HNO₃ and NH₃ over wet particles.*

**Q6.** Molar fraction of nitrate over total of $HNO_3 + NO_3^-$ highly depends on particle pH (e.g., Nah et al., 2018). The authors attributed a strong positive correlation between particulate nitrate and aerosol liquid water to the feedback loop between them, but no particle pH effect has been discussed. Can you exclude the possibility that increased particle pH would lead to an increase in particulate nitrate?

Nah et al., Atmospheric Chemistry and Physics 18.15 (2018): 11471-11491.

*Good point. We agree with the reviewer that the higher pH of aerosol particles is favorable for the equilibrium of HNO₃ into the particle phase (Nah et al., 2018). Therefore, We calculated aerosol pH using ISORROPIA Ⅱ and found out that the aerosol pH is not the driving factor for particulate nitrate formation in the investigated case.*

*A detailed discussion has been added into the Sec. 3.2 (Line 318~330, clean manuscript) as shown below.*

*"Aerosol pH is also an important factor on the particulate nitrate formation, higher pH is favorable for the equilibrium of HNO₃ into the particle phase (Nah et al., 2018). pH of the fine aerosol particles was calculated by ISORROPIA Ⅱ (Fountoukis and Nenes, 2007) during the investigated period. The model was running in 'forward mode' with chemical composition of NR-PM1 ($NO_3^-$, $SO_4^{2-}$, $Cl^-$, $NH_4^+$) and gas precursors ($HNO_3$, $HCl$, $NH_3$) by GAC-IC as inputs. And the model was running in 'metastable mode', assuming no solid existed in the system. Generally, the fine aerosol particles became more acidic with pH dropping from ~8 down to ~4 when $NR$-$PM_1$ mass concentration increased from ~12 μg/m³ up to >300 μg/m³ as shown in Figure 5 and Figure 6. This declining trend of pH is not favorable for the HNO₃ partitioning into the particle phase*

*(Nah et al., 2018). However, a clear enhanced trend of molar ratio of particulate nitrate in the total nitrate as a function of NR-PM$_1$ mass concentration was observed correspondingly (as shown in Figure 5 and Figure 6). Therefore, in this case the increase of aerosol liquid water is more likely to be the driving factor of particulate nitrate formation compared to the influence of pH."*

[Figure]

*Figure 5 in the revised manuscript (revised from Figure 3 in the old version) The time series of chemical composition measured by HR-ToF-AMS (left axis), calculated aerosol pH by ISORROPIA II (inner right axis) and molar ratio of particulate nitrate in the total nitrate (gas+particle phase) shown on outer right axis.*

[Figure]

*Figure 6 in the revised manuscript. The pH of the fine aerosol particles (left axis) and the molar ratio of particulate nitrate in the total nitrate (gas+particle phase) (right axis) as a function of NR-PM₁ mass concentrations.*

**Q7.** If the mutual effect of both nitrate and aerosol particle liquid water is the determine factor for increasing nitrate, why did the mass concentration of ammonium nitrate reach a plateau at aerosol liquid water of $> 20$ $\mu g/m^3$?

*Thanks so much for your comment on this point. I made a mistake by using the $PM_{2.5}$ nitrate mass concentration from GAC-IC measurement as a function of $PM_1$ aerosol liquid water. The revised figure was shown in revised Figure 4 below, a clear co-increase of particulate nitrate and aerosol liquid water was observed, which has been shown in Figure 1.*

*The previous description (Line 319~322, original manuscript) has been revised in the manuscript, as shown below.*

*"Consistently, a significant co-increase of particulate nitrate and aerosol liquid water was observed during haze development as shown in Figure 4. At first, a steep increase*

*of particulate nitrate in total nitrate mass ratio (from ~12% to ~98%) was observed as the aerosol liquid water enhanced up to ~20 µg/m³. And then, the particulate nitrate mass kept increasing with further increase of aerosol liquid water. We observed that, ~98% of nitrate was present as particle phase when aerosol liquid water was higher than ~20 µg/m³."*

[Figure]

*Revised Figure 4 in the manuscript. The relationship between aerosol particle liquid water and the molar ratio of particulate nitrate in the total nitrate,* $\mathbf{m_{NH_4NO_3}/(m_{HNO_3}+m_{NH_4NO_3})}$ *(left axis) during the nighttime 18:00~07:00⁺¹ (green solid triangle) and the daytime at 07:00 ~ 18:00 (red solid triangle), and mass concentration of particulate nitrate as a function of aerosol liquid water (right axis) during the period of February 29 to March 5, 2016. Here, particulate nitrate was measured by HR-ToF-AMS and the HNO₃ in the gas phase was measured by GAC-IC. Aerosol liquid water was calculated by H-TDMA-derived method.*

**Q8.** Finally, an increase in RH also leads to increasing aerosol liquid water. As shown in Fig. 1, the increased nitrate is coincident with the elevated RH during the haze period.

How do the authors discriminate the mutual effect from this elevated RH effect?

*The aerosol liquid water is determined by the ambient RH and chemical composition. So, the variations of ambient RH has been included in the changes of aerosol liquid water. We demonstrated the increasing of ambient RH as a prerequisite of the mutual promotion effect between particulate nitrate and liquid water.*

**Minor comments:**

**Q1.** Lines 109-113: Is the discussion about $NO_3^-/SO_4^{2-}$ in China?

*This discussion is based on $NO_3^-/SO_4^{2-}$ in Beijing. This has been clarified in the manuscript.*

**Q2.** Line 123: Need to be more explicit about how the reactive uptake o precursors and the thermodynamic equilibrium of ammonium nitrate change. Enhancing or increasing?

*Here, changing represents enhancing. This sentence has been clarified in the update version as "This leads to an increased aerosol liquid water content (Wu et al., 2018), which will enhance the particulate nitrate formation by increasing the reactive uptake of precursors and the thermodynamic equilibrium of ammonium nitrate (Cheng et al., 2016;Wang et al., 2016;Wang et al., 2017;Yun et al., 2018;Yue et al., 2019)."*

**Q3.** Line 279: theoretical calculations of what?

*Here, the theoretical calculations of condensational loss rate of $N_2O_5$ and $HNO_3$. This has been clarified in the update version.*

**Q4.** Line 293: Please use either liquify or liquefy for consistency.

*Thanks for your kindness. This has already been corrected in the update version.*

**Q5.** Line 346: HNO3 instead of particulate nitrate?

*Thanks for your kindness. This has already been corrected in the update version.*

[revised manuscript text omitted]

Xie, Y., Wang, G., Wang, X., Chen, J., Chen, Y., Tang, G., Wang, L., Ge, S., Xue, G., Wang, Y., and Gao,

J.: Observation of nitrate dominant PM2.5 and particle pH elevation in urban Beijing during the winter of 2017, Atmos. Chem. Phys. Discuss., 2019, 1-25, 10.5194/acp-2019-541, 2019.

Xu, Q., Wang, S., Jiang, J., Bhattarai, N., Li, X., Chang, X., Qiu, X., Zheng, M., Hua, Y., and Hao, J.: Nitrate dominates the chemical composition of PM2.5 during haze event in Beijing, China, Science of The Total Environment, 689, 1293-1303, https://doi.org/10.1016/j.scitotenv.2019.06.294, 2019.

Zhang, R., Jing, J., Tao, J., Hsu, S. C., Wang, G., Cao, J., Lee, C. S. L., Zhu, L., Chen, Z., Zhao, Y., and Shen, Z.: Chemical characterization and source apportionment of PM2.5 in Beijing: seasonal perspective, Atmos. Chem. Phys., 13, 7053-7074, 10.5194/acp-13-7053-2013, 2013.

---

## Author Comment (AC3) · 2 Jan 2020

**Response to comments of anonymous referees # 2**

**General comments**

This study presented observation results and theoretical calculations, and thereby proposed the so-called "mutual promotion effect between aerosol particle liquid water and nitrate formation". This study focused on an interesting topic, and is presented in a relatively clear way. However, the major problem I see is the confusion between "equilibrium" and "formation". I would recommend this manuscript for publication in ACP only if the following concerns can be nicely addressed.

As described above, this study kind of mixed up the concept of "equilibrium" and "formation". Now let's assume a system without $N_2O_5$. If the aerosols are already deliquencent, as the RH increase, the liquid water content (LWC) will increase, and the gas-particle partitioning of both $NH_3$ and $HNO_3$ will be influenced. Indeed, the final result might be that more $HNO_3$ partitioned on the pariticle phase and increasing the LWC, until a new equilibrium is reached that with more LWC and higher particle-phase $NO_3^-$ fraction. However, this process should be viewed as a new "equilibrium" driven by the elevated RH, and not "formation" of $HNO_3$. In this sense, there's no so-called "mutual promotion formation" concept. A lot of factors could influence this equilibrim, including aerosol acidity, activity and phase partitioning, but in any sense, there's no

"formation" in terms of total $HNO_3$ in the system.

Moreover, the authors seems to argue that that although gas-phase $NH_3$ and $HNO_3$ are also "supersaturated" under dry conditioned (in comparison with the equilibrium dissociation constant of $NH_4NO_3$ (Kp) under dry conditions), their condensation loss is "limited by its uptake rate". Once the aerosols get deliquescent, the update rate would be accelerated and more $NO_3^-$ will be present onto the particle phase. I agree this process may happen during the initial dry-wet transition periods, but what's the timescale? I

would say that the system would transfer from the "rate-limited regime" under dry conditions to "equilibrium-controlled regime" soon after this transition, on the scale of hours. Afterwards, as I described above, the apparent increase in NO$_3$ is only a result of equilibrium moving due to RH variations, if there're no influences from other factors. In this sense, I don't get the where the "mutual promotion" lies in – if it's referring merely to those several hours, I won't consider it as an important process in the haze development.

*Many thanks to the reviewer for the valuable comments and suggestions. I agree with the reviewer that it is essential to interpret clearly about the "equilibrium" or "formation", "uptake rate limited" or "equilibrium-controlled regime", and discuss the relative contributions of HNO$_3$ partitioning and N$_2$O$_5$ hydrolysis to particulate nitrate. We therefore have re-written the Sec. 3.2. 1) We found out that the aerosol particles were not reaching equilibrium during 29$^{th}$ Feb-4$^{th}$ Mar., 2019 as shown in Figure 3 below, which means the gaseous NH$_3$ and HNO$_3$ was supersaturated in the atmosphere throughout this haze event. In this case, the mutual promotion of aerosol liquid water and particulate nitrate enhancement happened throughout the haze development, and the increasing RH accelerated this process. I do apologize that I didn't clearly describe the "formation", is in terms of particulate nitrate, sometimes I described as nitrate (can refer to total nitrate in the system as well) which might mislead the audience. I have revised the terminology as suggested to avoid the unnecessary confusion, details shown below. 2) We agree with the reviewer that the "rate-limited regime" could be just in the time scale of hours after particles get deliquescent and system would transfer to "equilibrium-controlled regime" soon with the presence of aerosol liquid water. The corresponding discussion of "limited by its uptake rate" has been removed.*

*We have revised the Sec. 3.2 of the manuscript, as shown below.*

*"To illustrate the facilitation of particulate nitrate enhancement from HNO$_3$ in the presence of liquid water, we performed the theoretical calculation of equilibrium*

*between particulate $NH_4NO_3$ and gaseous $NH_3$ and $HNO_3$ under dry and ambient conditions, respectively. The dissociation constant of $NH_4NO_3$ (Kp) in dry condition was calculated using Eq. [5] without considering the influence of the liquid water. As shown in Figure 3, the equilibrium Kp in the dry condition ranged from 0.06 (275.3 K) to 4.61 (291.5 K) $ppb^2$ during the 'haze period'. Taking account of the aerosol liquid water, the equilibrium vapor pressure of $HNO_3$ and $NH_3$ over particles was calculated by E-AIM Model Ⅱ (www.aim.env.uea.ac.uk). Note that this calculation assumes negligible interaction between dissolved organic components and the activity of $NO_3^-$. In the presence of aerosol associated water, the product of equilibrium vapor pressure of $NH_3$ and $HNO_3$ calculated from E-AIM was 10~60% lower than the equilibrium Kp in the dry condition during the marked 'haze period'. This means, the presence of aerosol liquid water changed the equilibrium and would favor the particulate nitrate enhancement. However, the aerosol particles didn't reach the equilibrium between particulate $NH_4NO_3$ and the gases ($NH_3$ + $HNO_3$) during the investigated period, as the measured product of the $NH_3$ and $HNO_3$ partial pressure (2.55~9.63 $ppb^2$) was supersaturated compared to the equilibrium values in both dry and deliquescent particles. In this case, the partitioning of gaseous $NH_3$ and $HNO_3$ in the atmosphere into the particle phase could be accelerated and leaded particulate nitrate enhancement as increasing of ambient RH. Owing to the nature of highly hydrophilic, the increased ammonium nitrate mass fraction leads to further water uptake. Such a mutual promotion of particulate nitrate and aerosol liquid water enhancement becomes more pronounced with the increasing pollution throughout the haze event owing to the simultaneously increasing ambient RH. Consistently, a significant co-increase of particulate nitrate and aerosol liquid water was observed during haze development as shown in Figure 4. At first, a steep increase of particulate nitrate in total nitrate mass ratio (from ~12% to ~98%) was observed as the aerosol liquid water enhanced up to ~20 $\mu g/m^3$. And then, the particulate nitrate mass kept increasing with further increase of aerosol liquid water. We observed that, ~98% of nitrate was present as particle phase when aerosol liquid water was higher than ~20 $\mu g/m^3$. It is worth noting that $N_2O_5$ hydrolysis during nighttime can contribute extra $HNO_3$ in the wet denuding method*

*within GAC-IC system. This effect might cause uncertainty on the molar ratio of particulate nitrate in the total nitrate as a function of aerosol liquid water during nighttime. However, the consistency of this function between daytime and the nighttime (Figure 4) suggests a negligible influence of $N_2O_5$ interference on our analysis during the investigated period."*

[Figure]

*Figure 3 in the old version. The comparison of the calculated temperature-dependent dissociation constant of $NH_4NO_3$ (Kp) (Seinfeld. and Pandis., 2006) in the absence of liquid water, the product of equilibrium vapor pressure of gaseous $NH_3$ and $HNO_3$ from E-AIM, and the product of mixing ratios of gaseous $NH_3$ and $HNO_3$ measured by GAC-IC ($M\_pNH_3pHNO_3$). Here, Kp is colored by the ambient temperature ranging 265~293K during February 29 to March 5, 2016.*

The only way that I would agree the concept of "mutural promotion formation" is that, the uptake by $N_2O_5$ under high RH is so important that it contributed signicantly to the total $HNO_3$. This is not shown in this study. I would suggest to distinguish the relative contribution of $HNO_3$ partitioning and $N_2O_5$ uptake on the final enhancement of particle-phase $NO_3^-$.

*3) A previous study, as part of this BEST-ONE campaign, has distinguished the relative contribution of $HNO_3$ (75~99%) and $N_2O_5$ (1~25%) to the particulate nitrate (Lu et al., 2019), as shown in Fig. R1.*

*The relative contribution of particulate nitrate from N₂O₅ and HNO₃ has been added to the beginning of the Sec. 3.2 Mutual promotion between liquid water and particulate nitrate enhancement as shown below.*

*"Lu et al. (2019) conducted a box model to calculate the potential particulate nitrate formation during the same investigated period of the BEST-ONE project. They found out that HNO₃ from daytime photooxidation of NO₂ was the major source of the particulate nitrate (>75%), whereas the contribution of N₂O₅ pathway was lower than 25% (Lu et al., 2019)."*

[Figure]

*Fig. R1. Calculated contributions from HNO₃ (·OH+NO₂) shown as dashed red line and total particulate nitrate formation potential (solid red line) by box model during the same observational period. The difference between these two was the N₂O₅ contribution. This figure is sourced from the Figure S13 of Lu et al. (2019).*

Lu, K., Fuchs, H., Hofzumahaus, A., Tan, Z., Wang, H., Zhang, L., Schmitt, S. H., Rohrer, F., Bohn, B., Broch, S., Dong, H., Gkatzelis, G. I., Hohaus, T., Holland, F., Li, X., Liu, Y., Liu, Y., Ma, X., Novelli, A., Schlag, P., Shao, M., Wu, Y., Wu, Z., Zeng, L., Hu, M., Kiendler-Scharr, A., Wahner, A., and Zhang, Y.: Fast Photochemistry in Wintertime Haze: Consequences for Pollution Mitigation Strategies, Environmental Science & Technology, 53, 10676-10684, 10.1021/acs.est.9b02422, 2019.

Seinfeld., J. H., and Pandis., S. N.: Atmospheric Chemistry and Physics: from air pollution to climate change, John wiley & Sons, INC, 2006.